# Nurses’ Awareness and Actual Nursing Practice Situation of Stroke Care in Acute Stroke Units: A Japanese Cross-Sectional Web-Based Questionnaire Survey

**DOI:** 10.3390/ijerph182312800

**Published:** 2021-12-04

**Authors:** Yukari Hisaka, Hirokazu Ito, Yuko Yasuhara, Kensaku Takase, Tetsuya Tanioka, Rozzano Locsin

**Affiliations:** 1Department of Adult Nursing, Gifu University Graduate School of Medicine, Gifu 501-1194, Japan; hisaka@gifu-u.ac.jp; 2Graduate School of Health Sciences, Tokushima University, Tokushima 770-8509, Japan; 3Institute of Biomedical Sciences, Tokushima University, Tokushima 770-8509, Japan; h.itoh@tokushima-u.ac.jp (H.I.); yasuhara@tokushima-u.ac.jp (Y.Y.); locsin@tokushima-u.ac.jp (R.L.); 4Department of Rehabilitation, Anan Medical Center, Tokushima 774-0045, Japan; k.takase@ananmc.jp

**Keywords:** stroke care, nursing practice, nurse specialist, awareness, Japan, questionnaire survey

## Abstract

The awareness of care provided by stroke care unit (SCU) nurses in Japan to patients with an acute cerebrovascular accident (CVA) and the characteristic differences in their actual nursing practice were evaluated. A cross-sectional web-based questionnaire survey was administered to 1040 SCU nurses. Data collection and reporting procedures followed the STROBE Statement Checklist for cross-sectional studies. Exploratory factor analysis, using 52 observation items, identified eight factors with a factor loading > 0.4. For all factors, the actual practice was significantly lower than the awareness of the importance of nursing care for patients with acute CVA. Awareness and actual practice of recognition of patients’ physical changes (RPPCs) were high. The actual practice of RPPCs and preventing the worsening of acute stroke and related symptoms varied, depending on years of experience in acute phase stroke care. RPPCs in actual practice had a significantly higher score among certified nurses or certified nurse specialists. Their awareness of the importance of collaborating with therapists was low. On-the-job training can improve nurses’ competence and prevent worsening conditions in patients with CVA. An emphasis on enhancing practice experience toward patients with acute CVA and facilitating the deployment of certified nurses in SCUs can improve nursing care practice.

## 1. Introduction

Until 2019, cerebrovascular accident (CVA) was the second most common cause of death globally for the previous 20 years [1]. In 2014, the total number of patients with CVA in Japan was approximately 1,115,000 [2], and CVA was the fourth leading cause of death in 2020 [3]. It occurs suddenly, and acute CVA can lead to death or dysfunction [4]. CVA occurs because of cerebral hemorrhage caused by the rupture of blood vessels in the brain or because of brain infarction caused by blockage of blood vessels, thereby inhibiting cerebral blood flow. CVA results in cranial nerve conditions because of alterations in cerebral blood flow and compression of the brain parenchyma [5]. The healthy life expectancy in Japan is 10.2 years lower than the average life expectancy [1]. In addition, CVA is the most common underlying disease requiring severe-level care certification for long-term care insurance in Japan [6].

The incidence of CVA in Japan is 1.5–2 times higher than that in other developed Western countries [7], which indicates the need for sophisticated health care for patients with CVA. In Japan, a stroke care unit (SCU) is a specialized intensive care unit for patients with an acute CVA (e.g., cerebral infarction, cerebral hemorrhage, and subarachnoid hemorrhage). In an SCU, a specialized team of experienced physicians, nurses, and rehabilitation staff with expertise in CVA, intensively treats patients with CVA 24 hours a day from the early stage of onset. These patients require prompt diagnosis and proper treatment because the sooner they are treated, the more likely they are to return to their daily lives without sequelae. Treatment and rehabilitation starting from the acute phase by a team of medical experts have been reported to reduce mortality and increase the rate of functional recovery at discharge [8]. In addition, the care by professional development therapists and nurses has been reported to improve dysfunctions in the activities of daily living (ADL) in patients [9]. Previous studies have shown that discharge support provided from the acute stage of CVA reduces the length of hospital stays and improves ADL and the quality of life; this support consequently reduces mortality and the required level of assistance [10,11,12,13,14]. The results of a survey by Leys et al. [15] showed the necessity of having nurses specialized in stroke care in comprehensive CVA centers and primary stroke centers. A Japanese survey by Uehara [16] also showed similar results, which indicated the need for nurses’ specialization in stroke care in primary stroke centers.

Theofanidis and Gibbon [17] conducted a systematic review and found that advanced nursing care for a wide range of specific nursing interventions, such as continence management, pressure area care, swallowing management, early mobilization, pulmonary thromboembolism interventions, and early antiplatelet therapy is critical to facilitate patients’ early recovery. However, no study has focused on the care of patients with acute CVA in Japan. Furthermore, little is known about nurses’ awareness of observation items and methods for nursing care practice for patients with acute CVA in Japan.

Tulek et al. [18] conducted a survey of nurses in stroke care in 11 European countries. Nurses gave the following responses: within the first 48 hours after CVA onset, 95% monitored patients regularly; 94% started mobilization of the patients 24 hours after the patients were stable; 89% assessed patients’ ability to swallow; 73% followed a change in position of immobile patients, and 85% measured post-void residual urine volume. However, improvement was needed for staff education (70%), education for patients/families/careers (55%), and individual care plans for secondary prevention (62%). Other studies have focused on specific nursing skills, which included nursing roles and functions in acute and subacute rehabilitation care [19,20], standardized nursing intervention model for immobile patients with CVA [21], and early dysphagia screening to reduce the pneumonia rate [22].

Since 2006, an increasing number of hospitals in Japan have established SCUs. To provide competent professional nursing practice, the ratio of patients to nurses in SCUs is three patients to one or more nurses [23]. Nevertheless, only a few studies have been conducted on the status and effects of the care of patients with acute CVA in hospitals and SCUs, except for hospitals’ internal reports on nursing practice [24].

In the United Kingdom, the role of advanced nurse practitioners in acute stroke care has been studied [25]. The effects of Japan’s 2009 introduction of stroke rehabilitation nursing professional development for certified nurses (CNs) to become “stroke nurses” have not been fully examined [26]. Therefore, studying the extent of awareness for the care of patients with acute CVA in Japan and, especially, examining actual nursing practice is important. Furthermore, the relationships between the actual care of patients with acute CVA and the basic characteristics of nurses, such as years of experience caring for patients with acute CVA, and qualifications, such as nurses at an advanced level (e.g., CN and certified nurse specialist [CNS]) were found to be important. The elucidation of these questions may lead to the development of strategies to prepare nurses to deliver a certain level of nursing care for patients with acute CVA. A survey on the size of the facility and the number of beds in an SCU may lead to the development of on-the-job training (OJT) methodologies and measures to maintain and improve the quality of care for patients with acute CVA. This study aimed to determine the awareness of care provided to patients with acute CVA by SCU nurses in Japan and the characteristic differences in their actual nursing practice.

## 2. Materials and Methods 

### 2.1. Design and Method of Survey

Data collection and reporting were conducted based on the Strengthening the Reporting of Observational Studies in Epidemiology (STROBE) Statement Checklist for cross-sectional studies. The Survey Monkey platform (Momentive, Inc., San Mateo, CA, USA) was used for this cross-sectional web-based questionnaire survey in Japan. As of 2021, 181 medical facilities in Japan have SCUs. The population of SCU nurses was calculated to be 2700 nurses (approximately 15 nurses in each SCU). The nurses who work in SCUs include those who are dedicated to the SCU and those who also work in other units.

The participants in this study were recruited using the following procedure. The researcher provided a letter of invitation to participate and information about the study using the Survey Monkey URL to the directors of nursing, who agreed to distribute the document containing the URL to facilitate participants’ access to the survey instrument. The request letter included the study purpose, research collaboration procedure, ethical considerations, a URL, and a QR code access to the survey website.

Directors of nursing from 181 medical facilities with SCUs were requested to participate in this study. Fifty-three medical facilities agreed to participate in this study.

### 2.2. Participants

We invited 1040 nurses working in SCUs in Japan to participate in this study. However, only 850 (81.7%) nurses participated, of which 706 completed the questionnaire. The remaining participants were excluded on account of missing data fields.

### 2.3. Survey Period

The survey was conducted between February 2021 and March 2021.

### 2.4. Development of the Questionnaires

The questionnaire contents were developed using the following procedure:The contents of the curriculum for nursing practice for patients with acute CVA [27,28] were identified and used as questions about the care of patients with acute CVA. We also examined the contents of the systematic review conducted by Theofanidis and Gibbon [17] on nursing interventions in acute stroke care and identified and developed 52 questions. The questions were validated by three CNs in stroke rehabilitation nursing with 10 years of experience, a neurosurgeon and professor, and five faculty members of nursing colleges.The 52 questions on the awareness of the importance of nursing care in patients with acute CVA were compiled into a questionnaire format. The participants were asked to evaluate their awareness of nursing practice (i.e., the degree of importance) on the Likert scale: 1 = “not at all important”, 2 = “slightly important”, 3 = “neutral”, 4 = “somewhat important”, and 5 = “very important”. The participants evaluated their actual practice status using the Likert scale: 1 = “almost never”, 2 = “sometimes”, 3 = “neutral”, 4 = “very often”, and 5 = “almost always”. Please refer to Appendix A to review all items.The participants’ characteristics were age, sex, position (i.e., nurse manager or not in managerial position), certification status as a nurse or a nurse specialist, total years of experience as a nurse, total years of experience in the acute phase CVA care, number of beds in their hospitals, number of beds in the SCU, and the number of patients with CVA admitted in 1 year within 3 days of CVA onset.

### 2.5. Ethical Considerations

Ethical approval was obtained from the Ethics Committee of Gifu University (Gifu, Japan; approval number: 2020-228). Information on the study was made available on a web link that was accessible to the survey participants. It included an informed consent form to obtain permission to participate in the study and to collect the participants’ personal data. Participation was voluntary, and the participants could withdraw from the study at any time without any penalty. The Survey Monkey link was customized such that participants could only respond once. The consent form was provided through the link before the survey questions and was only accessible if the participants agreed to participate. To ensure confidentiality, identifying information, such as birthdates and contact information, which included email addresses, was not collected. In addition, personal information was kept confidential by securing access to the researcher’s computer with a password known only to the main researcher. Participants were presented with a reward of 300 Japanese yen after the completion of the survey, as approved by the ethics review board.

### 2.6. Statistical Analysis

The demographic data of the participants were evaluated using descriptive statistics. Before conducting factor analysis, the Kaiser–Meyer–Olkin (KMO) and Bartlett’s test of sphericity were conducted to evaluate the sampling adequacy and suitability of the data [29]. The acceptable value of the KMO test was ≥ 0.5 [30]. After the KMO test, exploratory factor analysis (EFA) was conducted using principal axis factoring and oblique rotation, which included direct Oblimin and Promax rotation with Kaiser normalization. The EFA used the 52 questions about the awareness of the need for the nursing of patients with acute CVA. Factors were determined using at least three questions with loadings greater than 0.4 [31]. In the first factor analysis, the number of factors from the eigenvalue, scree plot, and cumulative contribution rate were determined. Cronbach’s alpha was calculated to construct the reliability of the scales. The acceptable value of Cronbach’s alpha ranges from 0.70 to 0.95, which indicates internal consistency [32].

The mean of the factor points—derived from the total score for factors divided by the number of questions—was also calculated. The paired t-test was conducted to compare the awareness of the need for nursing for patients with acute CVA versus its actual practice. Welch’s t-test was used to determine the differences between the two samples: certification status and the number of beds in the SCU. Welch’s analysis of variance (ANOVA) with the Tamhane post hoc tests was used to conduct the following analysis for three or more groups: number of hospital beds and years of experience taking care of patients with acute CVA. Two statisticians evaluated the analysis framework and statistical processing results to ensure the validity of the analysis results. A *p*-value less than 0.01 was considered statistically significant. Statistical analyses were conducted using IBM SPSS Statistics (version 27.0; IBM Corp., Armonk, NY, USA).

## 3. Results

### 3.1. Characteristics of the Participants

As shown in Table 1, 706 participants completed the questionnaire. The ≤29-year-old age group comprised 44.2% of the participants. Of all participants, 89.4% were female. Nurses with ≤3 years of experience accounted for 21.0% of the participants, whereas nurses with ≤3 years of experience in acute phase CVA care accounted for 47.2% of the participants. With regard to the professional designation, general nurses accounted for 90.5% of the participants; CNs and CNSs in Japan, who were expert nurses in a specialized area, accounted for 5.1% of the participants. Twenty-four (66.7%) participants were CNs in stroke rehabilitation nursing. Their medical facilities were classified, based on the number of hospital beds, as follows: hospitals with 20–99 beds (8.8%), hospitals with 100–399 beds (42.1%), hospitals with 400–699 beds (32.9%), and hospitals with ≥700 beds (16.3%). With regard to the number of beds in the SCU, 63.6% of hospitals had 1–9 beds. Hospitals with 300–499 beds accounted for 29.0% of the annual number of patients with CVA admitted within 3 days of onset.

### 3.2. Results of the EFA

EFA was conducted using the 52 questions about nursing care for patients with acute CVA. After excluding 13 questions with a factor loading of <0.4, factor analysis identified eight factors by using 39 questions. The factors were named as follows:Factor 1: “Reacquisition of ADL” refers to the nursing practices that help to re-establish the ADL after its reduction by CVA.Factor 2: “Reduction of mental and social distress in patients and their families” refers to the nursing practice of being aware of the psychological distress of patients and family members and providing care.Factor 3: “Recognition of patients’ physical changes” (RPPCs) refers to the nursing practice of being aware of changes in the general condition and neurologic symptoms of patients after CVA and promptly reporting them to physicians.Factor 4: “Reduction of the risk of recurrence and providing discharge support” refers to the nursing practice of managing a smooth hospital discharge and transfer and preventing a post-discharge recurrence of CVA.Factor 5: “Collaboration with therapists” refers to the nursing practice of promoting patient training by physical therapists, occupational therapists, and speech therapists, and functional recovery training by nurses.Factor 6: “Reduction of patients’ physical distress” refers to the nursing practice of being aware of physical changes after CVA care and treatment-related physical pain and taking measures to reduce it.Factor 7: “Prevention of the worsening of acute CVA and related symptoms” refers to the nursing practice of preventing sudden changes in circulatory dynamics, respiratory complications, and an increase in intracranial pressure.Factor 8: “Appropriate management of patients’ physical conditions” refers to the nursing practice of preventing secondary complications related to the restriction of movement and promoting appropriate nutritional and fluid intake and safe medical treatment without harming patients’ health.

The overall Cronbach’s alpha coefficient was 0.95. The Cronbach’s alpha coefficient for each factor was as follows: Factor 1, α = 0.87; Factor 2, α = 0.90; Factor 3, α = 0.78; Factor 4, α = 0.86; Factor 5, α = 0.83; Factor 6, α = 0.84; Factor 7, α = 0.73; and Factor 8, α = 0.77 (Table 2).

### 3.3. Mean Difference between Awareness and the Actual Practice of Care for Patients with Acute CVA for Each of the Eight Factors

For all eight factors, scores for actual nursing care practice were significantly lower than the awareness of the importance of nursing care for patients with acute CVA (*p* < 0.001). The mean factor point for the awareness of the importance of nursing care for Factor 3 (i.e., “recognition of patients’ physical changes”) was highest (4.95 ± 0.15), whereas that for Factor 5 (i.e., “collaboration with therapists”) was lowest (4.57 ± 0.48). The mean score for nursing practice for Factor 3 (“recognition of patients’ physical changes”) was highest (4.71 ± 0.39), whereas that for Factor 2 (i.e., “reduction of mental and social distress in patients and their families”) was lowest (4.00 ± 0.75) (*p* < 0.001, respectively) (Table 3).

### 3.4. Results of the Comparisons of the Care of Patients with Acute CVA and Different Certifications of SCU Nurses

Table 4 shows the results of comparisons of the actual care of patients with acute CVA based on certification status as a nurse or a nurse specialist and the number of beds in the SCU. The Factor 3 scores for certification status of CNS was significantly higher (t = 2.90, *p* < 0.01). Comparisons of SCU groups, based on the number of beds revealed that Factor 5 scores for the 1–9 bed group were significantly higher (t = 2.75, *p* < 0.01).

### 3.5. Results of Comparisons of the Actual Care of Patients and the Capacity of Hospitals

Table 5 shows the results of the Welch’s ANOVA, which was used to compare the actual care of patients with acute CVA and the number of beds in their facilities and the years of nursing experience in providing acute phase CVA care. Comparisons of the actual care of patients with acute CVA, based on the number of beds in their facilities, showed a significantly higher total score for hospitals with 400–699 beds and hospitals with ≥700 beds than for hospitals with 20–99 beds (*p* < 0.01). Likewise, the results showed a significantly higher total score for hospitals with ≥ 700 beds than for hospitals with 100–399 beds (*p* < 0.01). The results also showed a significantly higher total score for hospitals with 400–699 beds than for hospitals with 100–399 beds (*p* < 0.01). For Factor 1, we found a significantly higher score for hospitals with 400–699 beds and hospitals with ≥700 beds than for hospitals with 20–99 beds (*p* < 0.01). For Factor 4, we found a significantly higher score for hospitals with ≥700 beds than for hospitals with 100–399 beds (*p* < 0.001). For Factor 5, we found a significantly higher score for hospitals with 400–699 beds and hospitals with ≥700 beds than for hospitals with 20–99 beds (*p* < 0.01).

For Factor 3, comparisons of the actual care of patients with acute CVA, based on years of experience in providing care for acute phase CVA patients, showed a significantly higher score in the group with 11–20 years of experience as a nurse than in the group with 0–3 years of experience as a nurse and the group with 6–10 years of experience as a nurse (*p* < 0.001). Furthermore, the results showed a significantly higher score in the group with 6–10 years of experience as a nurse than in the group with 0–3 years of experience as a nurse (*p* < 0.01). For Factor 7, the results showed a significantly higher score in the group with 11–20 years of experience as a nurse than in the group with 0–3 years of experience as a nurse (*p* < 0.001).

## 4. Discussion

This study aimed to determine the awareness of care provided by the SCU nurses to patients with acute CVA in Japan and the characteristic differences in their actual nursing practice. Based on a survey questionnaire, this study found that the awareness and actual practice of RPPCs were high. The actual practice of RPPCs and preventing the worsening of acute stroke and its related symptoms varied, depending on the number of years of experience in acute phase stroke care, the RPPCs in actual practice had a significantly higher score among certified nurses or certified nurse specialists; however, their awareness of the importance of collaborating with therapists was low. This study collected data from nearly 32% of SCU nurses in Japan. 

### 4.1. Characteristics of Participants

CN and CNS (qualified in Japan as stroke nurse specialists in a specialized area) accounted for 5.1% of the participants in this study; this was higher than the reported proportion of CNs and CNSs among the entire nurse population in Japan of 2.18% in 2020 [33]. Furthermore, 24 (66.7%) participants were CNs in stroke rehabilitation nursing, which is relatively high, compared to 3.25% of all CNs and CNSs in stroke rehabilitation nursing in Japan. CNs must have ≥5 years of clinical experience, ≥3 years’ experience of taking care of patients with acute CVA, 600 hours of specialized education in stroke nursing, and pass the Japanese Nursing Association examination. After such advanced training, nurses can become a CN in stroke rehabilitation nursing. Thus, it was considered that most of them have worked in an SCU where advanced level nursing for patients with CVA is required.

The number of beds in the hospitals in this study, compared to the number of beds in the medical care institutions in a dynamic survey by the Ministry of Health, Labour and Welfare of Japan in 2019 [34], was relatively high (hospitals with 20–99 beds, 8.8% vs. 35.5%; hospitals with 100–399 beds, 42.1% vs. 55.1%; hospitals with 400–699 beds, 32.9% vs. 7.85%; and hospitals with ≥ 700 beds, 16.3% vs. 1.5%), which indicated that most participants of this study worked in large hospitals. The reason may be that SCUs require the deployment of specialists and nurses and specialized equipment, such as MRI scanners; therefore, an SCU is usually built for the hospital. SCU hospitals with 1–9 beds account for 63.6% of all SCU hospitals in Japan, which indicates that many SCU hospitals have a smaller bed capacity than do intensive care unit hospitals. Twenty-nine percent of SCU hospitals admit 300–499 patients with CVA per year within 3 days of onset, which indicates that most SCU hospitals admit approximately one CVA patient per day.

### 4.2. EFA

EFA was conducted using 52 questions about the nursing care of patients with acute CVA. After excluding 13 questions with a factor loading of <0.4, factor analysis using 39 questions identified eight factors. The KMO sample adequacy was 0.939, and Bartlett’s test of sphericity was *p* < 0.001. The acceptable value of the KMO test was ≥ 0.5. Cronbach’s alpha coefficient ranged from 0.73 to 0.90, which indicated internal consistency.

In addition, this study compared the contents of the eight factors extracted in this study with the contents of the factors extracted from the literature review conducted by Theofanidis and Gibbon [17] on nursing interventions in acute stroke care delivery. Continence management was included in Factor 1 nursing practice (i.e., reacquisition of ADL) to remove unnecessary ureteral catheters at an early stage and to support patients to pass stool in the toilet instead of a diaper or on the bed. Pressure area care was included in Factor 7 nursing practice (i.e., prevention of the worsening of acute CVA and related symptoms) to provide care to prevent an increase in intracranial pressure. The management of swallowing was included in Factor 8 nursing practice (i.e., appropriate management of patients’ physical conditions) to provide care to support appropriate nutritional and fluid intake. Early mobilization was included in Factor 5 (i.e., collaboration with therapists) to coordinate and promote training by therapists and to provide training by nurses from the early phase of CVA. The prevention of pulmonary thromboembolism and early antiplatelet therapy were included in Factor 8 nursing practice (i.e., appropriate management of patients’ physical conditions) to prevent deep vein thrombosis (i.e., secondary complications related to restriction of movement).

We believe that nearly all details of nursing interventions in acute stroke care delivery reviewed by Theofanidis and Gibbon [17] were covered by the eight factors identified in this study. Furthermore, the results showed an awareness among nurses regarding the importance of Factor 2 (i.e., reduction of mental and social distress in patients and their families), Factor 6 (i.e., reduction of patients’ physical distress), and Factor 4 (i.e., reduction of the risk of recurrence and providing discharge support) in acute phase stroke care in Japan.

In Japan, nursing practice for patients with acute stroke emphasizes mental support for patients immediately after CVA onset, social support including patients’ families, and follow-up after discharge to avoid CVA recurrence. This may be because of the access to specialized nursing textbooks for patients with CVA and training on the stroke cure and care guidelines published by the Japan Stroke Society in 2015 [35].

### 4.3. The Mean Difference between Awareness and the Actual Practice of Care for Patients with Acute CVA for Each of the Eight Factors

After analyzing the results for all factors, the score for the actual nursing care practice for patients with acute stroke was significantly lower than that of the awareness of the importance of nursing care for patients with acute stroke. However, based on self-answered questionnaires, the low practice scores may not necessarily indicate poor practice but the lack of nurses’ confidence in their nursing practice [36]. The low scores may also be attributed to the following factors: (1) functional recovery after a brain injury takes time; (2) special skills are required for the observation of and communication with CVA patients because their responses may be unclear owing to cognitive impairment and disturbance of consciousness and (3) because SCU patients are at risk of sudden worsening, nurses may fail in providing care to prevent worsening because of poor understanding of the physical changes in patients. In particular, nurses with less experience in acute phase stroke care may be unsure of their ability to provide appropriate care to patients with acute CVA. Additional obstacles include nurses’ busy schedules and CVA patients’ various neurological dysfunctions. Nurses may not be able to find sufficient time to provide appropriate care because understanding patient conditions and supporting their daily activities take time.

Thus, a career development strategy, such as high-quality education, is necessary for experienced and new neuroscience nurses [36]. Regarding the items with significant differences in the recognition and the need for nursing practice for patients with acute stroke, it was estimated that nurses may believe that they cannot practice nursing or may not be able to practice, even if they recognize its necessity. However, further investigation is necessary to determine if a difference exists between perception and nursing practice due to lack of confidence.

However, if nursing practice is not actually followed, clarifying the existence of a responsible physical factor, such as the work environment, staff shortage, or the individual ability of the nurse to improve the quality of nursing, is necessary. If the cause is the lack of self-confidence in nursing practice, we believe that in-service education is necessary to boost confidence.

Based on the factor-wise analysis of the mean scores, the awareness and actual practice of “RPPCs” were both high. The high Factor 3 score may be attributed to the following characteristics of patients with CVA [37]: (1) patients with acute CVA have a higher risk of recurrence and sudden changes in pathology [38,39]; (2) many CVA patients cannot describe their physical changes because of aphasia or disturbance of consciousness; (3) nurses have to monitor subtle intracranial changes from physical findings of patients; and (4) early treatment affects the subsequent prognosis of all patients with respect to recurrence or worsening. In addition, one reason that many nurses responded that they always provide nursing care for Factor 3 may be that observation items on physical findings are required as the tasks of standard nursing.

Factor 5 (i.e., “collaboration with therapists”) showed the lowest score for the awareness of the importance of nursing care. In the past, patients with acute CVA were believed to require rest during the acute and unstable periods. However, early ambulation and rehabilitation have positive effects on prognosis [40]. Previous research [8,41] indicates that cooperation with therapists and rehabilitation by therapists is important in the management of acute CVA. Furthermore, education on the importance of rehabilitation and nursing care in the management of acute CVA needs improvement.

Factor 2 (i.e., “reduction of mental and social distress in patients and their families”) showed the lowest mean score in nursing practice. The study period was from February 2021 to March 2021. One reason that many nurses responded that they have not been able to provide family nursing care and practice during this period may be because visits to inpatients were forbidden or restricted in many hospitals to prevent the spread of COVID-19 since April 2020 Family nursing care methods focusing on infection prevention and control in SCUs need to be established.

### 4.4. Comparative Analysis of the Care of Patients with Acute CVA and the Characteristics of SCU Nurses

The actual practice of RPPCs and “prevention of the worsening of acute CVA and related symptoms” varied by years of experience in acute phase CVA care. Moreover, RPPCs in actual practice had a significantly higher score for CNs and CNSs. The curriculum for CNs in CVA rehabilitation nursing may include “monitoring and care of CVA patients to prevent worsening,” which focuses on the pathology of CVA patients (e.g., awareness of physical changes in patients).

Advanced practice nurse and physician collaboration is a promising model for healthcare quality improvement for inpatient stroke care [42]. In addition, interprofessional teamwork supports quality stroke care and underpins clinical confidence. Competent nurses’ roles include having up-to-date knowledge about stroke treatments, sharing best practice guidelines, instilling clinical confidence, and demonstrating transformational leadership. The critical point is whether such competent nurses will be able to translate clinical confidence into autonomous clinical decision-making [25]. Based on this perspective, nursing practice for patients with acute CVA may be improved by increasing the proportion of CNs and CNSs among SCU nurses in the future.

On comparing SCU groups, based on the number of beds in the factor analysis, the one- to nine-bed group had a significantly higher score for Factor 5 (i.e., “collaboration with therapists”). This finding suggested that hospitals with nine beds or fewer promote collaboration between nurses and therapists. In 2018, the “Stroke and Cardiovascular Disease Control Act” was enacted in Japan to improve medical care for patients with CVA [43]. Therefore, the number of SCU beds will be expected to increase in the future, and caution must be taken to prevent a decrease in the quality of nursing practice.

With regard to the total score, based on the number of beds in their facilities, hospitals with large bed capacities had significantly higher scores. These hospitals may have more nurses, better nursing management, and OJT systems. Other reasons may be that large hospitals have better nurse employment conditions and accept more college graduates, which leads to more nurses who actively gain knowledge and skills. Hospitals with a large bed capacity had a significantly higher score for Factor 1 (i.e., “reacquisition of ADL”) and Factor 4 (i.e., “reduction of mental and social distress in patients and their families”).

Appropriate continuous nursing intervention is required to enhance the quality of life of patients with CVA [44]. Nursing practice for the acquisition of ADL and the prevention of CVA recurrence has recently been added to the curriculum for nurses in Japan. Therefore, large hospitals may have more nurses who receive basic nursing education.

Hospitals with a large bed capacity had a significantly higher score for Factor 5 (i.e., “collaboration with therapists”). Other reasons may be that large hospitals have a higher number of therapists and their SCUs have full-time therapists, and interdisciplinary rehabilitation conferences are conducted regularly. Thus, large hospitals having SCUs with a small number of beds may promote collaboration between nurses and therapists. By contrast, the unprecedented nursing practice suggested the necessity of OJT for rehabilitation in small hospitals.

The analysis based on years of experience in acute phase CVA care also revealed a significantly higher score for Factor 3 (i.e., “recognition of patients’ physical changes”) in the group with 6–20 years of nursing experience than in all groups with 0–3 years of nursing experience. Moreover, a significantly higher score for Factor 7 (i.e., “prevention of the worsening of acute CVA and related symptoms”) in the group with 11–20 years of nursing experience than that of all groups with 0–3 years of nursing experience may be because nursing practice for preventing CVA worsening is difficult, even for nurses with years of experience, if they lack experience in acute phase CVA care. For nurses to become confident about their nursing practice takes 11 to 20 years of experience in acute phase CVA care. The results of this study suggested that even experienced nurses require OJT to prevent the worsening of CVA if they lack experience in taking care of such patients.

The accumulation of experience in a specific area and experience as a nurse contributes to the improvement of nursing practice in that area [45,46]. An individual nurse’s education level and years of experience influence the level of expertise, although gains in the probability of an individual nurse being an expert can also be achieved through having a more educated nursing staff overall [47].

### 4.5. Limitations and Generalizability of This Study

Our study has numerous potential limitations that need to be considered when interpreting our findings. With regard to age, the ≤29-year-old age group accounted for 44.2% of the participants in this study. This percentage was higher than that of a survey in 2018 in which the ≤29-year-old group accounted for 21.2% of nurses in Japan [48]. In addition, 89.4% of the participants were female. Considering that 92.9% of nurses in Japan in 2018 were female, the proportion of male nurses in our study was comparatively higher [48]. Compared with the proportion of management positions in nursing in Japan of 14.3% in 2019 [49], the ratio of general nurses (90.5%) was high in this study. This study was a survey of nursing practice; therefore, the response rate of nurses in management positions was considered low.

Nurses with ≤3 years of experience accounted for 21.0% of the participants in this study, although those with ≥3 years of experience in acute phase CVA care accounted for 47.2%. Most nurses with ≥3 years of general experience had ≤3 years of experience in acute phase CVA care. Thus, most SCU nurses were relatively less experienced in acute CVA care.

The discussions in this paper are based on nursing practices for patients with CVA in Japan; thus, the lack of knowledge regarding these practices internationally provides scope for future research. To the best of our knowledge, our study is the first that has gathered information on nurses’ awareness and actual nursing practice in acute stroke units in Japan. Thus, this study provides novel and important information, thereby advancing the research agenda on this topic.

## 5. Conclusions

This study aimed to determine the awareness of care provided to patients with acute CVA by SCU nurses in Japan and the characteristic differences in their actual nursing practice. For all factors, the actual practice was significantly different from the awareness of the importance of nursing care that nurses possessed for patients with acute CVA. The awareness and actual practice of “RPPCs” were both more. However, nurses’ awareness of the importance of “collaboration with therapists” was less. RPPCs in actual practice had a significantly higher score for CNs and CNSs. Therefore, enhancing the practice experience of patients with acute CVA and facilitating the deployment of CNs in SCUs can improve nursing care practice. Concerning the number of beds in facilities, actual practice scores were higher in hospitals with large-scale facilities. The results suggest the necessity of OJT in small hospitals, especially for “reacquisition of ADL,” “reduction of the risk of recurrence and providing discharge support,” and “collaboration with therapists.” Concerning the number of beds in SCU, actual practice scores were higher in hospitals with ≤9 beds for “collaboration with therapists.” The actual practice of “RPPCs” and “prevention of the worsening of acute CVA and related symptoms” varied, based on years of experience in acute phase CVA care. Therefore, for nurses who have little experience in caring for patients in the acute phase of CVA, providing OJT is necessary to avoid aggravation and to understand the physical changes peculiar to the acute phase of CVA.

## Figures and Tables

**Table 1 ijerph-18-12800-t001:** Participants’ demographic characteristics (*N* = 706).

Characteristic	Frequency (*N*)	Percentage (%)
**Age (years)**		
20–29	312	44.2
30–39	179	25.4
40–49	157	22.2
≥50	58	8.2
**Sex**		
Female	631	89.4
Male	75	10.6
**Years of experience as a nurse**		
0–3	148	21.0
4–5	105	14.9
6–10	173	24.5
11–20	167	23.7
≥21	113	16.0
**Years of experience taking care of patients with acute CVA**		
0–3	333	47.2
4–5	28	4.0
6–10	124	17.6
11–20	145	20.5
≥21	76	10.8
**Position**		
Nurse manager	67	9.5
Staff nurse	639	90.5
**Certification status as a nurse specialist**		
Certified Nurse or Certified Nurse Specialist	36	5.1
General nurse	670	94.9
**Number of hospital beds**		
20–99	62	8.8
100–399 beds	297	42.1
400–699 beds	232	32.9
≥700 beds	115	16.3
**Number of beds in the SCU**		
1–9	449	63.6
≥10 beds	257	36.4
**Number of patients with CVA admitted in 1 year within 3 days of onset**		
99 patients and hereinafter	47	6.7
100–199	177	25.1
200–299	168	23.8
300–499	205	29.0
≥500 patients	109	15.4

**Table 2 ijerph-18-12800-t002:** Results of exploratory factor analysis on the awareness of the need for nursing for patients with acute CVA.

Description of the Factors (The Overall Cronbach’s Alpha Coefficient Is 0.95)		Factor Loading							
		F1	F2	F3	F4	F5	F6	F7	F8
Factor 1: Reacquisition of ADL (Cronbach’s alpha coefficient = 0.87)									
Q33	Should recognize assistance needs and the levels of ADL in patients with acute stroke	**0.86**	0.04	0.01	0.00	0.10	0.09	0.07	0.09
Q34	Should detect dysfunction-affecting ADL in patients with acute stroke	**0.83**	0.03	0.00	0.00	0.09	0.03	0.04	0.08
Q35	Should provide nursing care to help patients with acute stroke to regain their ADL	**0.69**	0.10	0.03	0.06	0.00	0.03	0.04	0.10
Q31	Should provide nursing care to patients with acute stroke for better sleep and rest	**0.55**	0.04	0.03	0.02	0.10	0.11	0.07	0.06
Q32	Should provide nursing care to improve consciousness disorder in patients with acute stroke	**0.54**	0.16	0.15	0.13	0.03	0.02	0.26	0.01
Q30	Should provide nursing care to patients with acute stroke to promote their recovery	**0.52**	0.07	0.03	0.06	0.23	0.11	0.12	0.11
Q36	Should help patients with acute stroke to perform ADL by themselves	**0.49**	0.18	0.08	0.04	0.07	0.09	0.03	0.19
Q46	Should recognize the medical history and lifestyle risk factors for the onset of stroke in patients with acute stroke	**0.43**	0.01	0.10	0.24	0.01	0.20	0.00	0.18
**Factor 2: Reduction of mental and social distress in patients and their families (Cronbach’s alpha coefficient = 0.90)**									
Q44	Should provide nursing care for family-related mental distress in patients with acute stroke	0.00	**0.86**	0.02	0.13	0.00	0.11	0.08	0.11
Q40	Should recognize the need for family support in patients with acute stroke	0.06	**0.81**	0.05	0.00	0.02	0.06	0.03	0.07
Q45	Should provide nursing care for the need for family support in patients with acute stroke	0.07	**0.75**	0.01	0.00	0.10	0.03	0.12	0.14
Q39	Should recognize the mental distress in the family of patients with acute stroke	0.03	**0.70**	0.03	0.06	0.04	0.11	0.05	0.04
Q38	Should try to recognize the social distress in patients with acute stroke	0.30	**0.54**	0.08	0.02	0.02	0.04	0.06	0.00
**Factor 3: Recognition of patients’ physical changes (Cronbach’s alpha coefficient = 0.78)**									
Q5	Should recognize the changes in consciousness disorder due to stroke	0.07	0.11	**0.81**	0.03	0.12	0.00	0.02	0.13
Q13	Should report changes in the disease state of patients with acute stroke to physicians at the appropriate time	0.14	0.04	**0.61**	0.03	0.07	0.02	0.06	0.08
Q6	Should recognize changes in the motor dysfunction due to stroke	0.16	0.02	**0.60**	0.01	0.03	0.16	0.00	0.03
Q11	Should recognize the changes in the general condition of patients with acute stroke	0.02	0.00	**0.53**	0.02	0.01	0.00	0.14	0.06
Q3	Should recognize the need for treatment in patients with acute stroke	0.12	0.15	**0.47**	0.02	0.12	0.01	0.05	0.06
Q4	Should recognize the changes in intracranial hypertension due to stroke	0.09	0.11	**0.46**	0.02	0.00	0.08	0.04	0.09
**Factor 4: Reduction of the risk of recurrence and providing discharge support (Cronbach’s alpha coefficient = 0.86)**									
Q48	Should teach patients with acute stroke about lifestyle changes after hospital discharge to avoid the risk of recurrence	0.09	0.09	0.02	**0.94**	0.02	0.16	0.10	0.07
Q49	Should provide nursing care and guidance to patients with acute stroke (and their families if patient family support is needed in post discharge life) to avoid the risk of recurrence	0.05	0.16	0.04	**0.85**	0.01	0.17	0.02	0.01
Q47	Should explain the risk of recurrence to patients with acute stroke	0.18	0.03	0.05	**0.54**	0.04	0.01	0.07	0.06
Q51	Should provide nursing care to facilitate the transfer of patients with acute stroke to the hospital	0.10	0.18	0.05	**0.53**	0.00	0.21	0.06	0.13
Q52	Should provide nursing care to facilitate hospital discharge of patients with acute stroke	0.09	0.17	0.07	**0.52**	0.04	0.29	0.11	0.19
Q50	Should share the prognosis of patients with acute stroke with other healthcare providers	0.02	0.28	0.01	**0.44**	0.06	0.02	0.18	0.16
**Factor 5: Collaboration with therapists (Cronbach’s alpha coefficient = 0.83)**									
Q27	Should recognize the maximum physical ability of patients with acute stroke during training/exercise guided by the therapists	0.01	0.06	0.00	0.07	**0.84**	0.04	0.06	0.02
Q26	Should know the details of training/exercises for patients with acute stroke guided by the therapists	0.12	0.08	0.02	0.02	**0.77**	0.01	0.02	0.02
Q28	Should communicate to therapists about changes in patients with acute stroke that affect their training/exercise	0.13	0.10	0.07	0.04	**0.59**	0.10	0.04	0.01
Q29	Should facilitate not only therapist-guided training/exercise but also provide training/exercise by nurses	0.17	0.06	0.10	0.01	**0.58**	0.01	0.01	0.02
**Factor 6: Reduction of patients’ physical distress (Cronbach’s alpha coefficient = 0.84)**									
Q22	Should make attempts to reduce pain due to physical changes caused by a stroke	0.04	0.10	0.13	0.07	0.04	**0.66**	0.11	0.10
Q21	Should try to recognize the distress caused to patients with acute stroke as they are unable to communicate to others	0.01	0.17	0.09	0.08	0.04	**0.64**	0.04	0.02
Q23	Should provide nursing care to patients with acute stroke to minimize physical distress through treatment and care	0.04	0.08	0.07	0.05	0.01	**0.44**	0.25	0.12
**Factor 7: Prevention of the worsening of acute stroke and related symptoms (Cronbach‘s alpha coefficient = 0.73)**									
Q15	Should provide nursing care to prevent sudden changes in the circulatory dynamics of patients with acute stroke	0.05	0.23	0.03	0.04	0.05	0.08	**0.82**	0.01
Q14	Should provide nursing care to prevent exacerbation of intracranial hypertension in patients with acute stroke	0.16	0.14	0.20	0.01	0.03	0.02	**0.68**	0.13
Q16	Should provide nursing care to prevent respiratory complications in patients with acute stroke	0.01	0.09	0.09	0.03	0.06	0.11	**0.42**	0.23
**Factor 8: Appropriate management of patients’ physical conditions (Cronbach’s alpha coefficient = 0.77)**									
Q18	Should provide nursing care to patients with acute stroke to avoid the risk of secondary complications due to restricted movement	0.04	0.12	0.05	0.07	0.04	0.01	0.02	**0.70**
Q19	Should provide nursing care to ensure optimal nutrition and fluid intake in patients with acute stroke	0.07	0.02	0.03	0.01	0.10	0.09	0.05	**0.64**
Q20	Should provide nursing care to patients with acute stroke to avoid the risk of physical injury and to ensure safe medical treatment	0.02	0.04	0.15	0.10	0.09	0.15	0.06	**0.45**
Q17	Should ensure that patients with acute stroke receive appropriate treatment from physicians	0.03	0.07	0.31	0.02	0.14	0.03	0.19	**0.41**
**Factor Correlation Matrix**									
F1		1.00							
F2		0.71	1.00						
F3		0.43	0.26	1.00					
F4		0.67	0.64	0.36	1.00				
F5		0.65	0.63	0.28	0.50	1.00			
F6		0.47	0.45	0.35	0.37	0.39	1.00		
F7		0.52	0.52	0.48	0.48	0.45	0.42	1.00	
F8		0.62	0.55	0.48	0.49	0.49	0.49	0.64	1.00

Note. *N* = 706. Exploratory factor analysis was conducted using principal axis factoring and oblique rotation, which included direct Oblimin and Promax rotations with Kaiser normalization. Factor loadings above 0.40 are shown in boldface text. The Kaiser–Meyer–Olkin sample adequacy was 0.939 and Bartlett’s test of sphericity was *p* < 0.001. F = Factor.

**Table 3 ijerph-18-12800-t003:** Results of paired *t*-test comparing the awareness of the need for nursing for patients with acute CVA with its actual practice.

	Awareness		Actual practice		*t*	*p*
Variable	M	SD	M	SD		
The mean value of the total score divided by the number of questions	4.80	0.25	4.38	0.45	26.37	***
F1: Reacquisition of ADL	4.80	0.31	4.43	0.51	19.84	***
F2: Reduction of mental and social distress in patients and their families	4.69	0.45	4.00	0.75	25.88	***
F3: Recognition of patients’ physical changes	4.95	0.15	4.71	0.39	16.19	***
F4: Reduction of the risk of recurrence and requirement of discharge support	4.81	0.34	4.18	0.71	24.96	***
F5: Collaboration with therapists	4.57	0.48	4.13	0.66	18.27	***
F6: Reduction of patients’ physical distress	4.85	0.32	4.48	0.56	17.93	***
F7: Prevention of the worsening of acute stroke and related symptoms	4.90	0.26	4.58	0.53	15.91	***
F8: Appropriate management of patients’ physical conditions	4.86	0.27	4.61	0.47	14.24	***

Note. *N* = 706. The paired sample t-test was used to compare awareness and actual practice. The mean factor point is derived from the total score for factor divided by number of questions. F = Factor, M = mean, SD = standard deviation, *t* = *t*-value, *p = p*-value. *** *p* < 0.001.

**Table 4 ijerph-18-12800-t004:** Welch’s *t*-test between the two samples for actual practice.

Certification status	a: Certified Nurse or Certified Nurse Specialist (*n* = 36)		b: General Nurse (*n* = 670)		*t*	*p*
	M	SD	M	SD		
Total	4.54	0.43	4.37	0.45	0.39	0.70
F1: Reacquisition of ADL	4.56	0.53	4.42	0.51	1.61	0.11
F2: Reduction of mental and social distress in patients and their families	4.10	0.70	4.00	0.76	0.79	0.43
F3: Recognition of patients’ physical changes	4.85	0.29	4.71	0.39	2.90	**
F4: Reduction of the risk of recurrence and requirement of discharge support	4.40	0.56	4.17	0.71	1.96	0.05
F5: Collaboration with therapists	4.29	0.56	4.12	0.66	1.49	0.14
F6: Reduction of patients’ physical distress	4.62	0.50	4.48	0.56	1.50	0.13
F7: Prevention of the worsening of acute stroke and related symptoms	4.78	0.48	4.57	0.53	2.50	0.02
F8: Appropriate management of patients’ physical conditions	4.77	0.39	4.60	0.47	2.52	0.02
**Number of beds in the SCU**	**a: 1–9** **(*n* = 449)**		**b: ≥10 beds** **(*n* = 257)**		** *t* **	** *p* **
Total	4.41	0.44	4.33	0.47	2.37	0.02
F1: Reacquisition of ADL	4.45	0.50	4.38	0.53	1.97	0.05
F2: Reduction of mental and social distress in patients and their families	4.05	0.72	3.92	0.81	2.15	0.03
F3: Recognition of patients’ physical changes	4.74	0.35	4.66	0.44	2.49	0.01
F4: Reduction of the risk of recurrence and requirement of discharge support	4.21	0.69	4.13	0.74	1.50	0.13
F5: Collaboration with therapists	4.18	0.61	4.04	0.73	2.75	**
F6: Reduction of patients’ physical distress	4.50	0.55	4.46	0.57	0.91	0.36
F7: Prevention of the worsening of acute stroke and related symptoms	4.59	0.50	4.56	0.58	0.85	0.40
F8: Appropriate management of patients’ physical conditions	4.63	0.44	4.58	0.50	1.33	0.18

Note. *N* = 706. Welch’s t-test was used to compare the two samples.F = factor. M = mean, SD = standard deviation, *t* = *t*-value, *p* = *p*-value. ** *p* < 0.01.

**Table 5 ijerph-18-12800-t005:** Welch’s analysis of variance with Tamhane post hoc tests for actual practice.

Number of Hospital Beds	a: 20–99 (*n* = 62)		b: 100–399 (*n* = 297)		c: 400–699 (*n* = 232)		d: ≥700 Beds (*n* = 115)		*F*	*p*	Multiple Comparisons		
	M	SD	M	SD	M	SD	M	SD					
Total	4.18	0.55	4.34	0.45	4.45	0.43	4.48	0.35	9.00	***	a < c **, a < d **, b < d **		
F1: Reacquisition of ADL	4.19	0.65	4.38	0.52	4.49	0.47	4.53	0.42	8.08	***	a < c **, a < d **		
F2: Reduction of mental and social distress in patients and their families	3.77	0.92	3.93	0.78	4.12	0.70	4.07	0.67	5.16	**			
F3: Recognition of patients’ physical changes	4.61	0.56	4.69	0.39	4.74	0.38	4.79	0.29	3.69	0.01			
F4: Reduction of the risk of recurrence and requirement of discharge support	3.91	0.84	4.08	0.73	4.28	0.65	4.35	0.60	9.20	***	b < d ***, a < c **, a < d **, b < c **		
F5: Collaboration with therapists	3.84	0.70	4.10	0.67	4.20	0.63	4.23	0.60	6.17	***	a < c **, a < d **		
F6: Reduction of patients’ physical distress	4.35	0.67	4.46	0.57	4.53	0.52	4.52	0.53	1.85	0.14			
F7: Prevention of the worsening of acute stroke and related symptoms	4.43	0.63	4.55	0.56	4.61	0.49	4.68	0.45	3.57	0.01			
F8: Appropriate management of patients’ physical conditions	4.46	0.54	4.58	0.49	4.64	0.44	4.70	0.40	4.48	**			
**Years of experience in taking care of patients with acute stroke**	**a: 0–3** **(*n* = 333)**		**b: 4–5** **(*n* = 28)**		**c: 6–10** **(*n* = 124)**		**d: 11–20** **(*n* = 145)**		**e: ≥21 years** **(*n* = 76)**		** *F* **	** *p* **	**Multiple Comparisons**
Total	4.35	0.45	4.41	0.45	4.43	0.41	4.42	0.46	4.35	0.56	1.08	0.36	
F1: Reacquisition of ADL	4.40	0.52	4.45	0.47	4.46	0.46	4.42	0.57	4.41	0.66	0.40	0.81	
F2: Reduction of mental and social distress in patients and their families	4.00	0.75	4.05	0.77	4.02	0.77	3.95	0.74	3.88	0.76	0.43	0.78	
F3: Recognition of patients’ physical changes	4.66	0.43	4.71	0.36	4.78	0.32	4.83	0.30	4.74	0.46	4.64	**	a < d ***, a < c **
F4: Reduction of the risk of recurrence and requirement of discharge support	4.13	0.74	4.20	0.72	4.24	0.62	4.23	0.68	4.20	0.77	0.82	0.51	
F5: Collaboration with therapists	4.12	0.64	4.19	0.65	4.19	0.62	4.06	0.76	3.96	0.85	1.23	0.30	
F6: Reduction of patients’ physical distress	4.47	0.55	4.48	0.59	4.49	0.53	4.54	0.54	4.45	0.72	0.21	0.93	
F7: Prevention of the worsening of acute stroke and related symptoms	4.50	0.56	4.61	0.48	4.65	0.52	4.74	0.43	4.58	0.66	4.17	**	a < d ***
F8: Appropriate management of patients’ physical conditions	4.57	0.49	4.65	0.45	4.62	0.45	4.72	0.39	4.60	0.53	2.10	0.08	

Note. *N* = 706. Welch’s analysis of variance with Tamhane post hoc tests was also used to conduct the analysis for three groups or more. F = factor, M = mean, SD = standard deviation, *p* = *p*-value. ** *p* < 0.01 and *** *p* < 0.001.

## Data Availability

The data are not publicly available due to privacy and ethical restrictions.

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
