# Peer review of "Nurses’ Awareness and Actual Nursing Practice Situation of Stroke Care in Acute Stroke Units: A Japanese Cross-Sectional Web-Based Questionnaire Survey"

_ijerph, 2021, doi:10.3390/ijerph182312800_

Round 1

Reviewer 1 Report

Thank you for asking me to review this article. The issue under consideration is very important and current, especially in relation to the maintenance of high quality standards considered indispensable in care practices. Therefore, analyzing the knowledge, attitudes and perceptions of health professionals is important in order to implement strategies aimed at improving the quality of care.
The premises are well described and the results are very interesting, however the work needs a thorough review before its publication.
In this regard, my suggestions are as follows:

- line 96, the authors describe that the questionnaire was administered through the Survey Monkey platform but they do not give any information about the choice of this tool compared to others, they refer to the functionality or the expected methodology of its use. It is suggested to deepen this section by clarifying the method of administration;
- line 99 the authors declare “The participants were 1040 nurses working in SCUs in Japan”. It is more accurate to say that "1040 nurses were invited to participate in the study". On line 171 it should be emphasized that of the 1040 invited to participate only 850 actually participated (81.7%) of which 706 completed the questionnaire entirely.
- For the purposes of reproducibility of the authors' experience in other contexts, the authors may think of inserting the questionnaire in the same form in which it was administered to healthcare personnel. In this regard, it is suggested to consult the papers doi: 10.3390 / ijerph17093185 and doi: doi: 10.12927 / hcpol.2020.26290. Given the organization of the contents, the proposed questions could be included in the method section in a simple explanatory table.
- lines 329-341 describe the inclusion of the various determinants within the factors considered. This section, as described, could be moved to the results paragraph.
- lines 351 and 352 are almost superimposable to lines 433 and 434; moreover, many of the contents in this section are described as results leaving little room for final considerations.
Finally, it is suggested to broaden the empirical value of the results obtained, that is the contribution they offer to theoretical and practical progress aimed at reducing the gap between awareness and nursing practice in relation to the care of patients in acute stroke units.

Author Response

 Author's Reply to Reviewer

Thank you for asking me to review this article. The issue under consideration is very important and current, especially in relation to the maintenance of high-quality standards considered indispensable in care practices. Therefore, analyzing the knowledge, attitudes and perceptions of health professionals is important in order to implement strategies aimed at improving the quality of care. The premises are well described, and the results are very interesting, however the work needs a thorough review before its publication. In this regard, my suggestions are as follows:

Response: We sincerely thank you and the reviewers for your thoughtful suggestions and insights, which have enriched the manuscript and produced a better and more balanced account of the research. The manuscript has been rechecked and appropriate changes have been made in accordance with the reviewers’ suggestions. The responses to their comments have been prepared. The revised sentences are presented in red font in the manuscript. We hope that the revised manuscript is now suitable for publication in your journal.

  • - line 96, the authors describe that the questionnaire was administered through the Survey Monkey platform, but they do not give any information about the choice of this tool compared to others, they refer to the functionality or the expected methodology of its use. It is suggested to deepen this section by clarifying the method of administration;

Response: Thank you for pointing this out. We have provided detailed information to clarify the method of administration and edited this section according to your knowledgeable insight (Lines 105-116, page 3).

  • - line 99 the authors declare “The participants were 1040 nurses working in SCUs in Japan”. It is more accurate to say that "1040 nurses were invited to participate in the study". On line 171 it should be emphasized that of the 1040 invited to participate only 850 actually participated (81.7%) of which 706 completed the questionnaire entirely.

Response: We have revised the sentences, based on your suggestions, as follows (Lines 120-122, page 3): “We invited 1,040 nurses working in SCUs in Japan to participate in this study. However, only 850 (81.7%) nurses participated, of which 706 completed the questionnaire.”

  • - For the purposes of reproducibility of the authors' experience in other contexts, the authors may think of inserting the questionnaire in the same form in which it was administered to healthcare personnel. In this regard, it is suggested to consult the papers doi: 10.3390 / ijerph17093185 and doi: doi: 10.12927 / hcpol.2020.26290. Given the organization of the contents, the proposed questions could be included in the method section in a simple explanatory table.

Response: We appreciate your review. We have read the papers you presented, and we agree with your opinion that this is an important point of view when considering reproducibility. However, because Survey Monkey is charged monthly, no link exists to show this survey item. In addition, due to the limited space of this manuscript, we believed that inserting the questionnaire in the same format as used for healthcare professionals was not possible. Therefore, we have added all of the questionnaire items as an Appendix (Line 667-673, page 20-23).

  • - lines 329-341 describe the inclusion of the various determinants within the factors considered. This section, as described, could be moved to the results paragraph.

Response: We are grateful for your suggestion. This study is a comparative study of the factor analysis results of this study and the previous study by Theofanidis and Gibbon. Therefore, this was mentioned in the Discussion section, as follows (Lines 361-363, page 15): “We believe that nearly all details of nursing interventions in acute stroke care delivery reviewed by Theofanidis and Gibbon were covered by the eight factors identified in this study.”

  • - lines 351 and 352 are almost superimposable to lines 433 and 434; moreover, many of the contents in this section are described as results leaving little room for final considerations.

Response: Thank you for noticing this. We have revised the sentences based on your suggestions and added references to the texts of lines 351 and 352 that you indicated (Lines 375-379, page 15). We have also deleted the sentence in the Conclusion section (lines 433 and 434).

  • Finally, it is suggested to broaden the empirical value of the results obtained, that is the contribution they offer to theoretical and practical progress aimed at reducing the gap between awareness and nursing practice in relation to the care of patients in acute stroke units.

Response: Thank you for this comment. We have added the following sentences in the Discussion section (Lines 391-402, page 15): “Thus, a career development strategy, such as high-quality education, is necessary for experienced and new neuroscience nurses. Regarding the items with significant differences in the recognition and the need for nursing practice for patients with acute stroke, it was estimated that nurses may believe that they cannot practice nursing or may not be able to practice, even if they recognize its necessity. However, further investigation is necessary to determine if a difference exists between perception and nursing practice due to lack of confidence. However, if nursing practice is not actually followed, clarifying the existence of a responsible physical factor, such as the work environment, staff shortage, or the individual ability of the nurse to improve the quality of nursing, is necessary. If the cause is the lack of self-confidence in nursing practice, we believe that in-service education is necessary to boost confidence.”

Reviewer 2 Report

Thank you for the opportunity to review manuscript #ijerph-1417179 “Nurses 'Awareness and Situation of the Actual Nursing Practice of Stroke Treatment in Acute Stroke Units in Japan”. The aim of the study was to determine the awareness of care provided to acute stroke patients by nurses in stroke care units (UCS) in Japan and the characteristic differences in their actual nursing practice. This is an article with a relevant research object for nursing practice and that has consequences for patient care and continuing health education.

The study was analyzed under the prism of the STROBE protocol (Strengthening the Report of Observational Studies in Epidemiology). In this sense, a small change in the title is necessary for the method to be identified, as provided for in the protocol.

In the introduction, the authors present an overview of the case of stroke cases in Japan and emphasize nursing practice as essential for the maintenance of daily activities and increased quality of life. Furthermore, they point out the need for specialist nurses at all levels of care, including primary health care.

The materials and methods follow all the steps of the STROBE protocol, which guarantees the quality of the search and the reliability of the findings. It is important to clarify the characteristics of the questionnaire such as its international validation.

The presentation of results is clear, objective and allows the reader to reflect on the findings and the relationship between definitions. Discussion of data is impoverished and re-presents the results. This was built with only 7 references and could be better worked, using international experiences as a basis.

Aware of the quality of this research, I indicate my favorable opinion on the post-publication of the discussion.

Author Response

Author's Reply to Reviewer

Thank you for the opportunity to review manuscript #ijerph-1417179 “Nurses 'Awareness and Situation of the Actual Nursing Practice of Stroke Treatment in Acute Stroke Units in Japan”. The aim of the study was to determine the awareness of care provided to acute stroke patients by nurses in stroke care units (UCS) in Japan and the characteristic differences in their actual nursing practice. This is an article with a relevant research object for nursing practice and that has consequences for patient care and continuing health education.

Response: Thank you for your help in improving our manuscript. The revised sentences are presented in red font in the manuscript.

  • The study was analyzed under the prism of the STROBE protocol (Strengthening the Report of Observational Studies in Epidemiology). In this sense, a small change in the title is necessary for the method to be identified, as provided for in the protocol.

Response: Thank you for your advice. We have revised the title to “Stroke Care in Acute Stroke Units: A Japanese Cross-sectional Web-based Questionnaire Survey.”

  • The materials and methods follow all the steps of the STROBE protocol, which guarantees the quality of the search and the reliability of the findings. It is important to clarify the characteristics of the questionnaire such as its international validation.

Response: Thank you for your insightful review. Owing to the space limitations of this manuscript, we believed that inserting the questionnaire in the same format that was used for healthcare professionals was not possible. Therefore, we have added all of the questionnaire items in the Appendix (Lines 141-142, page 3 and Line 667-673, pages 20-23).

  • The presentation of results is clear, objective and allows the reader to reflect on the findings and the relationship between definitions. Discussion of data is impoverished and re-presents the results.

Response: Thank you for your comment. We have reviewed our manuscript and edited it according to the STROBE Statement Checklist.

  • This was built with only 7 references and could be better worked, using international experiences as a basis.

Response: Thank you for your advice. The discussions are based on the information in Japan, and the lack of consideration, based on international experience, is a limitation this study and provides scope for future research. We have cited 13 new literature reports (36-38, and 41-50) in the Discussion section and Revised the Limitations section (Lines 503-508 page 17).

  • Aware of the quality of this research, I indicate my favorable opinion on the post-publication of the discussion.

Response: We are grateful to you for reviewing and helping us in improving our manuscript.

Reviewer 3 Report

Dear Authors, 
Thank you for exploring this important concern in nursing practice. Overall, this paper is clearly written and well presented.

There were a couple of words that need to be more clearly articulated. The use of the term young nurses could be easily misconstrued and has connotations attached to it. I would suggest finding different terminology, as it seems as though the nurses were experienced although not in care of patients with CVA. Also, the use of the word stroke is very generic, consider using CVA or Cerebral event, clot or bleed. The definition used in the introduction is very limited. Consider if all patients on this unit are diagnosed Stroke or any Cerebral event? What is the patient demographic? Is there diversity amongst the "stroke" patients and/or nurses?

The use of the word training to describe specialist education is also concerning, consider changing this, consider professional development, education modules etc. 

Kind Regards 

Author Response

Author's Reply to Reviewer

 English language and style are fine/minor spell check required

Response: Thank you for this comment. We appreciate the suggestion and have resubmitted the manuscript for English language editing. The revised sentences are presented in red font in the manuscript.

Dear Authors, 
Thank you for exploring this important concern in nursing practice. Overall, this paper is clearly written and well presented.

  • There were a couple of words that need to be more clearly articulated. The use of the term young nurses could be easily misconstrued and has connotations attached to it. I would suggest finding different terminology, as it seems as though the nurses were experienced although not in care of patients with CVA.

Response: Thank you for this comment. We have revised the term “young nurse” to “nurses with less experience” (Line 385-386, page 15).

  • Also, the use of the word stroke is very generic, consider using CVA or Cerebral event, clot or bleed.

Response: Thank you for this comment. We changed the word “stroke” to “cerebrovascular accident” (“CVA”). The terms “SCU” and “stroke care” used in Japan were not changed. We have also used the word “stroke” in the questionnaire as necessary to improve clarity.

  • The definition used in the introduction is very limited. Consider if all patients on this unit are diagnosed Stroke or any Cerebral event?

Response: We appreciate that you have mentioned this. We have added sentences in the Introduction section concerning this issue, as follows (Lines 44-50, page 2): “In Japan, a stroke care unit (SCU) is a specialized intensive care unit for patients with an acute CVA (e.g., cerebral infarction, cerebral hemorrhage, and subarachnoid hemorrhage). In an SCU, a specialized team of experienced physicians, nurses, and rehabilitation staff with expertise in CVA, intensively treats patients with CVA 24 hours a day from the early stage of onset. These patients require prompt diagnosis and proper treatment because the sooner they are treated, the more likely they are to return to their daily lives without sequelae.”

  • What is the patient demographic?

Response: Thank you for your question. We have added sentences in the Introduction section concerning this issue, as follows (Line 32-34, page 1): "In 2014, the total number of patients with CVA in Japan was approximately 1,115,000, and CVA was the fourth leading cause of death in 2020.”

  • Is there diversity amongst the "stroke" patients and/or nurses?

Response: Thank you for your question. We have added sentences regarding diversity in the Introduction section (Lines 44-50, page 2) and in the Materials and Methods section (Lines 108-110, page 3).

  • The use of the word training to describe specialist education is also concerning, consider changing this, consider professional development, education modules etc.

Response: Thank you for this suggestion. We have changed the word “training” to “professional development.” However, on-the-job training (OJT) is a well-established word in Japanese literature. Therefore, we have retained the abbreviation “OJT.”

Round 2

Reviewer 1 Report

The authors answered all observations exhaustively and worked on the proposed suggestions carefully.
I believe that the work has significantly improved in form and content and that it can represent an interesting contribution to the scientific literature so I believe that it can be published in its current form